# Incorporating Spatial Information into Goal-Conditioned Hierarchical Reinforcement Learning via Graph Representations

**Shuyuan Zhang**                                                    *shuyuan.zhang@mail.mcgill.ca*
*School of Computer Science*
*McGill University, Mila*

**Zihan Wang**                                                      *zihan.wang5@mail.mcgill.ca*
*School of Computer Science*
*McGill University, Mila*

**Xiao-Wen Chang**                                                       *chang@cs.mcgill.ca*
*School of Computer Science*
*McGill University*

**Doina Precup**                                                        *dprecup@cs.mcgill.ca*
*School of Computer Science*
*McGill University,*
*Google Deepmind,*
*CIFAR Fellow*

**Reviewed on OpenReview:** *https://openreview.net/forum?id=a7Bx4s5gA8*

## Abstract

The integration of graphs with Goal-conditioned Hierarchical Reinforcement Learning (GCHRL) has recently gained attention, as intermediate goals (subgoals) can be effectively sampled from graphs that naturally represent the overall task structure in most RL tasks. However, existing approaches typically rely on domain-specific knowledge to construct these graphs, limiting their applicability to new tasks. Other graph-based approaches create graphs dynamically during exploration but struggle to fully utilize them, because they have problems passing the information in the graphs to newly visited states. Additionally, current GCHRL methods face challenges such as sample inefficiency and poor subgoal representation. This paper proposes a solution to these issues by developing a graph encoder-decoder to evaluate unseen states. Our proposed method, Graph-Guided sub-Goal representation Generation RL (G4RL), can be incorporated into any existing GCHRL method when operating in environments with primarily symmetric and reversible transitions to enhance performance across this class of problems. We show that the graph encoder-decoder can be effectively implemented using a network trained on the state graph generated during exploration. Empirical results indicate that leveraging high and low-level intrinsic rewards from the graph encoder-decoder significantly enhances the performance of state-of-the-art GCHRL approaches with an extra small computational cost in dense and sparse reward environments.

## 1 Introduction

Traditional reinforcement learning methods face great challenges when learning policies in environments with long time horizons and sparse rewards. To address these challenges, Hierarchical Reinforcement Learning (HRL) methods have been proposed to break problems into smaller, more manageable subproblems conducive

to more efficient learning. Previous works (Sutton et al., 1999; Kulkarni et al., 2016; Vezhnevets et al., 2017; Levy et al., 2017) have demonstrated HRL's capability in handling large, sparse-reward environments. Among HRL methods, Goal-Conditioned Hierarchical Reinforcement Learning (GCHRL) has attracted much attention due to its well-defined paradigm (Nachum et al., 2018b) and its resemblance to the human thinking process (Zhao et al., 2022; Hsu et al., 2024). Although GCHRL has shown superior performance compared to non-hierarchical methods in some scenarios, questions such as how to learn better subgoal representations and explore the state space more efficiently (Nachum et al., 2018a; Guo et al., 2021) remain unanswered.

GCHRL methods typically utilize two levels of agents as described in Nachum et al. (2018b). The high-level agent chooses the next target state based on its current state, while the low-level agent decides how to reach this target. Both agents face their own challenges. The high-level agent suffers from sample inefficiency in environments with a large state-action space, a problem also encountered by non-hierarchical methods, while the low-level agent is trained solely by the reward signals derived from the distances between the current state and subgoals in the representation space, making the reward signals highly susceptible to poor subgoal representations.

Previous works have tried to either enhance the efficiency of high-level exploration (Huang et al., 2019; Zhang et al., 2022) or find a good subgoal space to boost GCHRL's performance in complex environments (Wang et al., 2024). To the best of our knowledge, while these methods have made progress in their respective aspects, no prior work has attempted to address the efficiency of high-level exploration and the accuracy of low-level learning signals in a unified framework. Yet, such integration is crucial, as increasing the effectiveness of the high-level agent and the accuracy of the low-level agent within a cohesive framework can yield performance improvements that surpass simple additive effects.

Recently, combining graph theory with RL has become a trend in the community (Lee et al., 2022; Gieselmann & Pokorny, 2021) as graphs are inherently well-suited for representing the environment and task structure.

Previous work has focused on areas such as decision-making through graph search or traversal (Wan et al., 2021; Shang et al., 2019; Eysenbach et al., 2019), and as well as using graphs as world models (Zhang et al., 2021; Huang et al., 2019). Yet, many previous works rely on pre-crafted graphs, limiting their generalizability. Additionally, most existing works (Zhu et al., 2022; Hong et al., 2022) constructed graphs directly from the original state space, which cannot provide meaningful guidance when the agent encounters a state that is not represented in those graphs.

To utilize graph representations when a new state (node) is encountered, transitioning to these representations (Hamilton, 2020; Chen et al., 2020; Khoshraftar & An, 2024) through graph learning is a viable option. Recent studies have shown that employing graph representations for learning can improve underlying RL performance (Klissarov & Precup, 2020; Klissarov & Machado, 2023).

In this paper, we propose a novel approach that simultaneously addresses all of the aforementioned problems. Specifically, we construct a state graph based on visited states until the number of nodes reaches a threshold and after that we update the graph by adding and dropping nodes as new states are visited. Using this graph, we build a subgoal space through graph learning that leverages both state representation and spatial connectivity. By generating subgoal representations through graph learning, we ensure that they capture their relative positions in the decision chain, thereby forming a more effective subgoal space. In estimating the distance between the current state and the intended subgoal in the original space, we use the distance between their corresponding representations in the subgoal space. This distance is then used to calculate the intrinsic reward for curiosity-driven exploration, aiming to improve sample efficiency across both high-level and low-level agents.

The main contributions of this paper are as follows:

- We propose a novel architecture that employs a graph encoder-decoder to embed spatial information into subgoal representations, enabling the evaluation of newly visited states. This architecture can be integrated into any GCHRL algorithm operating in environments with primarily symmetric and reversible transitions, enhancing performance across this class of problems.

- We present a method for the online construction of the state graph as a graph-based world model (Ha & Schmidhuber, 2018; Zhang et al., 2021) for the HRL agent by sampling from trajectories.

- We use novelty-based auxiliary rewards (Simsek & Barto, 2006; Nehmzow et al., 2013) derived from subgoal representations to improve sample efficiency for both high-level and low-level agents.

We tested our approach on several environments (Todorov et al., 2012) to assess the significance of our experimental results. The findings indicate that our method can significantly enhance the performance of the underlying HRL approach in terms of both sample efficiency and success rate/cumulative reward.

## 2 Preliminaries

### 2.1 Markov Decision Processes

As the most common framework for modeling reinforcement learning scenarios, the Markov Decision Process (MDP) (Puterman, 2014) is introduced as a tuple $< \mathcal{S}, \mathcal{A}, P, R, \gamma >$, defined as follows: At each time step $t$, the agent observes the current state $s_t \in \mathcal{S}$ provided by the environment and chooses an action $a_t \in \mathcal{A}$ according to its internal policy $\pi(a_t|s_t)$, which specifies the probability of choosing action $a_t$ given state $s_t$. The action is then executed, and the interaction with the environment leads the agent to a new state $s_{t+1}$ according to a transition probability function $P(s_{t+1}|s_t, a_t)$ which is known only to the environment. Subsequently, the agent receives a reward $r_t$, determined by the reward function $R(s_t, a_t)$ that evaluates the action taken in the current state and is also only visible to the environment. The agent aims to learn an optimal policy $\pi$ to maximize the expected discounted cumulative reward $\mathbb{E}_\pi \left[ \sum_{t=0}^{T} \gamma^t r_t \right]$, where $\gamma$ (with $0 \le \gamma < 1$) is a pre-defined discount factor used to prioritize immediate rewards over distant future rewards, thereby ensuring that the total reward remains finite.

### 2.2 Goal-conditioned Hierarchical RL (GCHRL)

Goal-Conditioned Reinforcement Learning (GCRL) trains agents to achieve specific goals, which are the target states. The agent receives an additional goal input $g_t$ along with the state input $s_t$ and learns a policy $\pi(a_t|g_t, s_t)$ that aims to achieve this goal. Goals are represented explicitly in the input to the policy, guiding the agent's actions towards desired outcomes. The reward function is often goal-dependent, providing positive feedback when the agent successfully reaches the desired goal state.

To deal with large and complex environments, Goal-Conditioned Hierarchical Reinforcement Learning (GCHRL) (Nachum et al., 2018b; Zhang et al., 2022; Wang et al., 2024) decomposes the learning task into a hierarchy of smaller, more manageable sub-tasks. Typically, there are two levels of agents. At time step $t$, the high-level agent chooses a subgoal $g_t$, a representation of a target state, and assigns it to the low-level agent to achieve as part of the overall task. This choice is made by sampling $g_t$ from the high-level policy $\pi_h(g_t|\phi(s_t))$, where $\phi : s \mapsto \mathbb{R}^d$ is the state representation function which gives a condensed representation of the state.

Each state $s_t$ can be mapped to its subgoal feature $g(s_t)$ by a subgoal feature extractor. Note that $g(s_t)$ is not the same as $g_t$. The former, $g(s_t)$, is the learned subgoal feature of the current state $s_t$, while the latter, $g_t$, is the target state we aim to reach from the state $s_t$ in one step or multiple steps.

Given the subgoal $g_t$ sampled from the high-level policy $\pi_h(g_t|\phi(s_t))$ for the current time step $t$ and the state representation vector $\phi(s_t)$, a low-level agent executes action $a_t$ based on the low-level policy $\pi_l(a_t|\phi(s_t), g_t)$. The low-level agent is trained using the intrinsic reward signal $r_{\text{int}}(s_t, g_t, a_t, s_{t+1}) = -\|\phi(s_{t+1}) - g_t\|_2^2$ to encourage it to achieve the subgoal.

Both agents can be implemented by any policy-based methods, including those introduced in previous works on policy gradients such as Fujimoto et al. (2018); Haarnoja et al. (2018) and Schulman et al. (2017).

### 2.3 Graph and MDP

Graph is a generic data structure, which can model complex relations among objects in many real-world problems. A graph is defined as $\mathcal{G} = (\mathcal{V}, \mathcal{E})$, where $\mathcal{V} = \{1, 2, \ldots, N\}$ is the set of nodes and $\mathcal{E} = \{e_{ij}\}$ is the set of edges without self-loops. The adjacency matrix of $\mathcal{G}$ is denoted by $\boldsymbol{A} = (\boldsymbol{A}_{i,j}) \in \mathbb{R}^{N \times N}$ with $\boldsymbol{A}_{i,j} = 1$ if there is an edge between nodes $i$ and $j$, otherwise $\boldsymbol{A}_{i,j} = 0$. The adjacency matrix can be extended to a *weighted* adjacency matrix, where $\boldsymbol{A}_{i,j}$ is a weight of the edge $e_{ij}$.

In MDP, a node can represent a state, while the edge weights can model the transition probabilities or reachability statistics between states.

## 3 Methods

This section presents our framework, Graph-Guided subGoal representation Generation (G4RL). Our method reshapes the subgoal space utilizing a state graph to incorporate the relative spatial information of visited states.

One drawback of previous hierarchical reinforcement learning algorithms (Nachum et al., 2018b; Kim et al., 2021; Zhang et al., 2022; Luo et al., 2024) is that the Euclidean distance calculated in the original state representation space between the current state and the intended goal does not accurately reflect the true progress of the low-level agent, as there is rarely a straight path between the current state and the subgoal in the space. As a result, the low-level agent trained with such information may receive an inaccurate reward signal, thus impairing its performance. Another issue is that, without appropriate constraints, the high-level agent may propose subgoals that are too difficult to reach, wasting exploration steps on pursuing infeasible targets (Zhang et al., 2022). Our proposed method aims to mitigate both problems by calculating the distance in a subgoal representation space between subgoal representations given by a graph encoder-decoder. This graph encoder-decoder captures the actual connectivity between states, ensuring that the generated subgoal representations respect adjacency information.

### 3.1 State graph

To record the visited states and their connections, we maintain a state graph $\mathcal{G} = (\mathcal{V}, \mathcal{E})$ with a fixed number $N$ of nodes[1]. This graph is built and updated during training, with no pre-training using expert data or handcrafted process involved in its construction.

Each node is labelled by the corresponding state and for each node $s_t$, the corresponding state representation vector $\phi(s_t)$, which is also referred to as the node feature, is stored. Edges in the graph represent connectivity between states. The graph is constantly updated during exploration.

We choose the state graph to be undirected for mainly two reasons: **(1) Efficiency:** An undirected graph requires fewer resources for storage and computation compared to a directed graph, and **(2) Method compatibility:** Our method relies on defining a deterministic distance between state/subgoal representations for each node pair. This is straightforward in an undirected graph but becomes problematic in a directed setting, where distances can be asymmetric or undefined. Related details will be explained in Sections 3.2 and 3.3.

This choice is based on the assumption that G4RL is designed for environments with symmetrical and reversible dynamics. However, some of our experiments demonstrate that G4RL can still enhance performance in partially asymmetric environments; please refer to the experiments section for further details.

#### 3.1.1 Graph construction

The graph is initialized with $N$ empty nodes and no edges. The corresponding weighted adjacency matrix $\boldsymbol{A}$ is set to an $N \times N$ zero matrix. We perform the GCHRL exploration process using randomly initialized policy

---

[1]The number of training states for the graph encoder-decoder grows quadratically with $N$ because the adjacency weight matrix has $N^2$ elements. The choice of $N$ depends on the machine's capabilities.

$\pi_h$ and $\pi_l$. Once the agent encounters a state representation never seen before, that is, the representation is different from any state representations stored in the graph, as described in equation (1), it stores the state representation $\phi(s_t)$ as the node feature of an empty node in the graph and build an edge between this node and the node corresponds to the previous state:

$$\forall_{s_v \in \mathcal{V}}, \|\phi(s_t) - \phi(s_v)\|_2 > \epsilon_d, \tag{1}$$

$$\boldsymbol{A}_{\phi(s_t),\phi(s_{t-1})} = \boldsymbol{A}_{\phi(s_{t-1}),\phi(s_t)} = 1, \tag{2}$$

where $\epsilon_d$ is a hyper-parameter controlling the distance threshold between state representations. When the agent encounters a state $s_t$ with feature $\phi(s_t)$ that is similar to several node representations already stored in the graph, it finds the state whose representation is the closest to the current state feature:

$$s_v = \underset{s_u : \|\phi(s_t) - \phi(s_u)\|_2 \leq \epsilon_d}{\arg\min} \|\phi(s_t) - \phi(s_u)\|_2. \tag{3}$$

Then the node $s_v$ is relabeled as $s_t$ and the weight for the edge $(s_{t-1}, s_t)$ is updated as follows:

$$\boldsymbol{A}_{\phi(s_{t-1}),\phi(s_t)} = \boldsymbol{A}_{\phi(s_t),\phi(s_{t-1})} := \boldsymbol{A}_{\phi(s_{t-1}),\phi(s_t)} + 1. \tag{4}$$

Note that a large weight indicates more frequent transitions between the underlying states.

We have used the Euclidean norm to define the distance between feature vectors. Since some elements may contain more spatial information than others, one can use a weighted Euclidean norm to define the distance between state representations instead.

### 3.1.2 Graph updating

The graph has a fixed number of nodes. Suppose the graph is now full. When a new state $s_t$ is encountered, if $s_v$ from equation (3) exists, as before we relabel the node as $s_t$ and perform edge update as shown in equation (4); Otherwise, we replace the oldest state node in the graph with the current state node, delete all edges previously linked to that node, and create an edge $(\phi(s_{t-1}), \phi(s_t))$ with weight $\mathbf{A}_{\phi(s_{t-1}),\phi(s_t)} = \mathbf{A}_{\phi(s_t),\phi(s_{t-1})} = 1$. Alternatively, we could replace the state node that is most weakly connected to the other nodes–that is, the node with the lowest sum of edge weights.

### 3.2 Graph encoder-decoder

To enable the assignment of suitable subgoal representations to every possible state, including unseen ones, we use node representations and edges to train a graph encoder-decoder. The parameter updates of the graph encoder-decoder and the policies during policy training are performed alternately in each episode.

The encoder-decoder starts training after the graph is full and continues periodically after processing a few trajectories. Section 3.3 will show the details of the training schedule.

The encoder $\mathbf{E}$ maps every state representation $\phi(s)$ to a subgoal representation $g(s)$. We use a feed-forward network (FFN) with several layers as the encoder $\mathbf{E}$:

$$g(s) = \mathbf{E}(\phi(s)) = \text{FFN}(\phi(s)). \tag{5}$$

The weight parameters of the feed-forward network will be learned through training. The decoder $\mathbf{D}$ takes two subgoal representations as input and outputs the inner product of these two representations:

$$\mathbf{D}(g(s_u), g(s_v)) = g(s_u)^T g(s_v). \tag{6}$$

We choose dot-product similarity based on the assumption that the similarity between two nodes, such as the overlap in their local neighbourhoods, is well captured by the dot product of their embeddings. This assumption is supported by prior work in the graph embedding literature (Ahmed et al., 2013; Cao et al., 2015; Ou et al., 2016).

The aim is to use the encoder-decoder structure to predict node relations. Naturally we can use $\boldsymbol{A}_{\phi(s_u),\phi(s_v)}$ as a measure of the relation between nodes $\phi(s_u)$ and $\phi(s_v)$. But for the sake of numerical stability in the training process, we use $\boldsymbol{A}_{\phi(s_u),\phi(s_v)}/\max_{\phi(s_i),\phi(s_j)} \boldsymbol{A}_{\phi(s_i),\phi(s_j)}$ as a measure. Thus the loss function is defined as:

$$\mathcal{L} = \sum_{\phi(s_u),\phi(s_v)\in\mathcal{V}} \left[\mathbf{D}(g(s_u),g(s_v)) - \boldsymbol{A}_{\phi(s_u),\phi(s_v)}/\max_{\phi(s_i),\phi(s_j)} \boldsymbol{A}_{\phi(s_i),\phi(s_j)}\right]^2. \tag{7}$$

This loss function can enforce the subgoal representation provided by the encoder to respect neighbouring features in the graph.

Note that in each training phase of the graph encoder-decoder (except the first one), we use the values of the parameters obtained from the last training phase as the initial point, which helps save computation cost.

## 3.3 Adaptive training schedule of the graph encoder-decoder

The graph stores evolving data, including state representations as node features and connection information in the weighted adjacency matrix $\boldsymbol{A}$, which are continuously updated during online training. Since the graph structure and content change at varying rates across episodes, training the graph encoder-decoder at fixed intervals can cause several issues: (1) high variance in earlier episodes, where sparse or unstable graph data may lead to unreliable model updates; (2) data underutilization, where intermediate graph states are overwritten before being used for training; and (3) overfitting in later episodes, as the model repeatedly trains on increasingly redundant data. To address these issues, we introduce an adaptive training schedule for the graph encoder-decoder, described in the following paragraph.

There are two types of data changing in the graph: node replacement and edge update. We introduce a variable $c$ to track the weighted number of data changes. Since the replacement of nodes has a much higher impact on the data than the edge update, we add $N-1$ to $c$ if a node replacement occurs, and add 1 to $c$ if an edge update happens:

$$c = \begin{cases} c + N - 1, & \text{if a node replacement happens,} \\ c + 1, & \text{if an edge update happens.} \end{cases} \tag{8}$$

When this variable exceeds a certain value, specifically a tolerance $\beta$ multiplied by the total number of non-diagonal elements $N^2 - N$ in the matrix $\boldsymbol{A}$. we perform one gradient update for the graph encoder-decoder, and then we reset $c$ to 0.

## 3.4 Hierarchical agent with graph encoder-decoder

Our proposed method involves traditional goal-conditioned settings and a subgoal representation extractor implemented by a graph encoder-decoder.

The high-level policy $\pi_h(g_t|\phi(s_t))$ nominates a subgoal every $K$ steps and is trained using the external environmental reward $r_{\text{ext}}$.

The policy can be implemented by any policy-based RL algorithm that takes transition tuples $(s_t, g_t, a_t, r_t, s_{t+1}, g_{t+1})$ as input. To encourage it to propose a subgoal that is not too difficult to reach from the current state $s_t$ for more efficient exploration, we add an intrinsic term to the high-level reward, considering the distance between the subgoal features of $s_t$ and $g_t$ in the subgoal space:

$$r_h(s_t, g_t, s_{t+1}) = r_{\text{ext}} + r_{\text{int}} = r_{\text{ext}} + \alpha_h \cdot \mathbf{D}(\mathbf{E}(\phi(s_t)), \mathbf{E}(g_t)), \tag{9}$$

where $\alpha_h$ is a hyperparameter that controls the significance of the intrinsic term in the high-level reward.

The low-level policy $\pi_l(a_t|\phi(s_t), g_t)$, however, operates in the subgoal space. While it still takes $\phi(s_t)$ and $g_t$ as input and outputs an atomic action $a_t$, we compute the reward based on distances in the subgoal space:

$$r_l(s_t, g_t, a_t, s_{t+1}) = -\|\phi(s_{t+1}) - g_t\|^2 + \alpha_l \cdot \mathbf{D}(\mathbf{E}(\phi(s_{t+1})), \mathbf{E}(g_t)), \tag{10}$$

where $\alpha_l$ is a hyperparameter controlling the significance of the reward term in the low-level reward. By computing the intrinsic reward in the subgoal space rather than in the state space, the function provides high values when proposed subgoals are easy to reach from the current location and low values when subgoals are close in the original state space but difficult to reach from the current location. The low-level agent can also be any policy-based algorithm.

### 3.5 Balancing between speed and performance

Due to the excessive comparisons between the current state representation and the node features during graph updates, as well as the training cost of the graph encoder-decoder on a large graph, our experiments show that the GCHRL method, after incorporating our method, takes approximately twice as long as the original GCHRL method.

To reduce the additional cost, we can either decrease the frequency of sampling candidates for node features, train the graph encoder-decoder with a subset of all available training data, or do both.

For the sampling frequency, instead of comparing the state representation with node features in each time step, we do it in every $t_c$ time steps. This may significantly speed up our method while maintaining satisfactory performance.

To reduce the training data in each graph encoder-decoder training cycle, we randomly sample node pairs in the graph instead of using every node pair for training.

In the experiments section, we will present the training time and performance of G4RL with the above two techniques applied.

### 3.6 Algorithm: GCHRL + G4RL

The detailed description of how our proposed strategy, G4RL, can be incorporated into typical GCHRL algorithms is given in Appendix D.

## 4 Experiments

In this section, we empirically evaluate the effectiveness of integrating G4RL into existing GCHRL methods. The experiment results demonstrate substantial improvements in both convergence speed and overall success rates achieved by our proposed approach. Additionally, we provide empirical evidence that the generated state graph accurately represents the underlying structure and relationships within the task environments.

### 4.1 Environment settings

We used AntMaze, AntGather, AntPush, AntFall and AntMaze-Sparse environments from the GYM MuJoCo library (Todorov et al., 2012). The first four involve complex navigation and manipulation tasks performed by a simulated multi-armed robot, while AntMaze-Sparse presents a particularly challenging scenario due to sparse reward signals, providing feedback only upon reaching the goal. Note that AntPush and AntFall contain asymmetric transitions which lead to irreversible state changes that can be difficult to learn with an undirected graph. We added these environments to show G4RL's robustness in inherently asymmetric environments. For the state representation, we selected a subset of raw state dimensions that contains spatial information (e.g. coordinates and arm angles) to serve as the node representation for the graph encoder-decoder. We deliberately aligned our choice of environments with prior work to ensure a fair and consistent comparison between the backbone algorithms and their G4RL-augmented versions. This allowed us to use the same environments, hyperparameters, and codebases provided by the original studies. Our goal was to demonstrate that G4RL can consistently enhance the performance of these backbone algorithms under comparable conditions.

## 4.2 Experimental Comparisons

We incorporated G4RL in the following existing GCHRL methods:

- **HIRO** (Nachum et al., 2018b): This is the first method which describes how the Goal-conditioned information can be integrated into hierarchical agents.

- **HRAC** (Zhang et al., 2022): This method enhances the performance of HIRO by training an adjacency network that produces subgoals easier to reach from the current subgoal.

- **HESS** (Li et al., 2022): This method applies a regularization term on consecutive subgoal representations in each update to stabilize the representation across episodes.

- **HLPS** (Wang et al., 2024): This method applies the Gaussian process on subgoal representations for a smoother representation update.

In addition to comparing these four GCHRL-G4RL methods with their native counterparts, we also compared them with the following non-hierarchical method:

- **TD3** (Fujimoto et al., 2018): This is a well-known non-hierarchical policy-based method designed for continuous action spaces and we use it to implement both high- and low-level agents.

Although reward is a key metric of an agent's learning ability, for AntMaze and AntMaze-Sparse, we compare success rates of these methods instead of their rewards on the corresponding tasks. This is because higher rewards in AntMaze/AntMaze-Sparse do not necessarily indicate better performance; the agent may achieve high rewards without reaching the goal.

The learning curves of baseline methods and G4RL-applied versions are plotted in Figure 1 and 2. Note that all the curves reported in Section 4 are averages from 20 independent runs and they have been equally smoothed for better visualization.

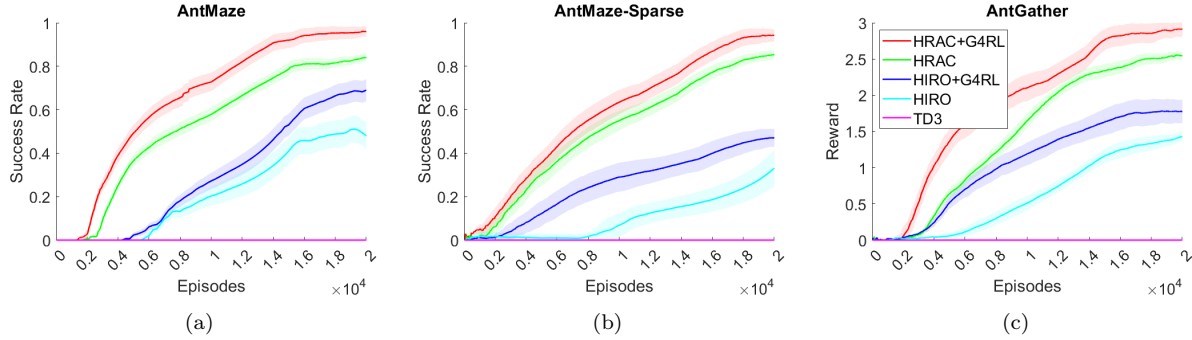

Figure 1: Success Rate on (a) AntMaze (b) AntMaze-Sparse and Reward on (c) AntGather, using HIRO, HIRO-G4RL, HRAC, HRAC-G4RL, and TD3. Incorporating G4RL in HIRO and HRAC significantly enhances their performance.

From Figures 1 and 2, we observe that, in all environments, incorporating G4RL in the base GCHRL methods significantly enhances their performance, further improving the already strong results of these hierarchical methods compared to the non-hierarchical method. Notably, G4RL-augmented methods not only achieve higher final success rates but also converge substantially faster, with the most significant improvements observed during the early stages of training.

To demonstrate the proposed method's effectiveness in environments with image-based state representations, we conducted experiments on AntMaze, AntPush, and AntFall, utilizing images as states, and compared the results with HESS and HLPS, along with their G4RL variations. We use Mean Squared Error (MSE)

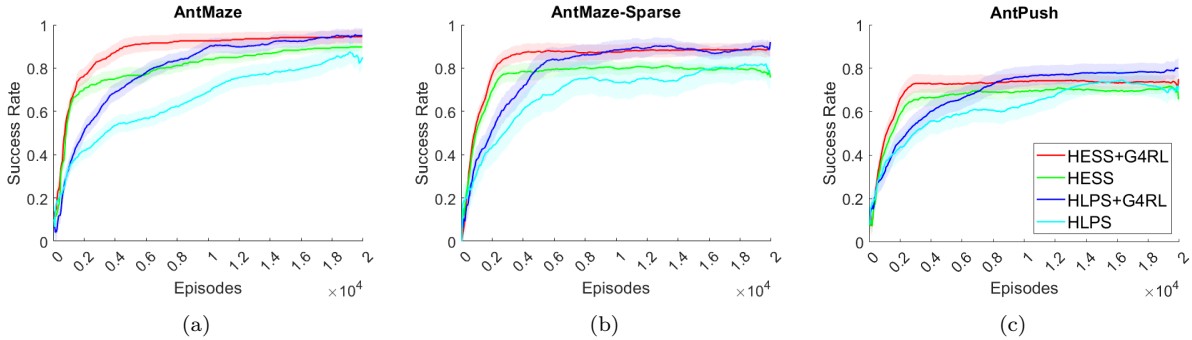

Figure 2: Success Rate on (a) AntMaze (b) AntMaze-Sparse and (c) AntPush, using HESS, HESS-G4RL, HLPS, HLPS-G4RL. Incorporating G4RL in HESS and HLPS significantly enhances their performance.

to measure the pixel-wise differences between image states to decide whether a new node should be added to the graph. The test results, given in Figure 3, show that methods incorporating G4RL exhibit faster convergence and achieve higher performance across all tested image-based environments.

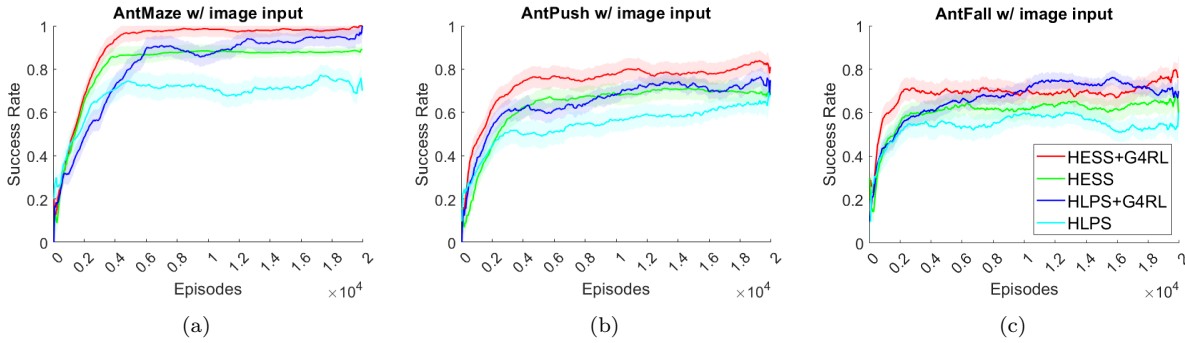

Figure 3: Success Rate on (a) AntMaze (b) AntPush and (c) AntFall with image state features, using HESS, HESS-G4RL, HLPS, HLPS-G4RL. Incorporating G4RL helps convergence and achieves higher performance across all tested image-based environments.

### 4.3 Ablation study

#### 4.3.1 The effect of high/low-level intrinsic reward

We consider the following variants of G4RL to show the effectiveness of adding high-level and low-level intrinsic rewards:

- **High+Low-level intrinsics**: Apply both equation (9) and equation (10) to the high-level and low-level rewards respectively.

- **High-level intrinsic only**: Apply equation (9) to the high-level rewards and set $\alpha_l = 0$ in equation (10) when it is applied to the low-level rewards.

- **Low-level intrinsic only**: Apply equation (10) to the low-level rewards and set $\alpha_h = 0$ in equation (9) when it is applied to the high-level rewards.

- **HIRO/HRAC/HESS/HLPS**: Vanilla baseline methods.

Same as before, all the curves reported in this section are drawn from results averaged across 20 independent runs. All curves have been equally smoothed for better visualization. Additionally, we have added the

un-smoothed versions of figures 4 and 5 in Appendix E to confirm that the conclusions are unaffected by the smoothing.

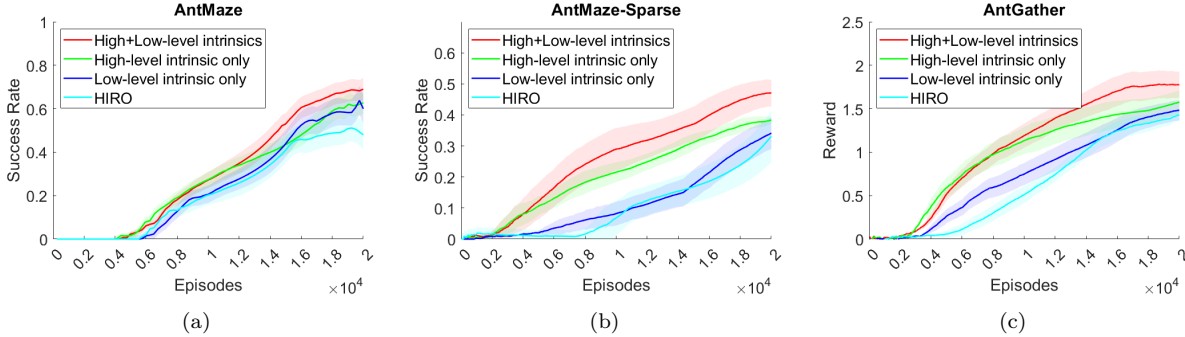

Figure 4: Success Rate on (a) AntMaze (b) AntMaze-Sparse and Reward on (c) AntGather using HIRO-G4RL, HIRO + High-level intrinsic, HIRO + Low-level intrinsic and HIRO. All curves have been equally smoothed for better visualization. The combination of high-level and low-level intrinsic rewards results in the highest success rates and fastest convergence.

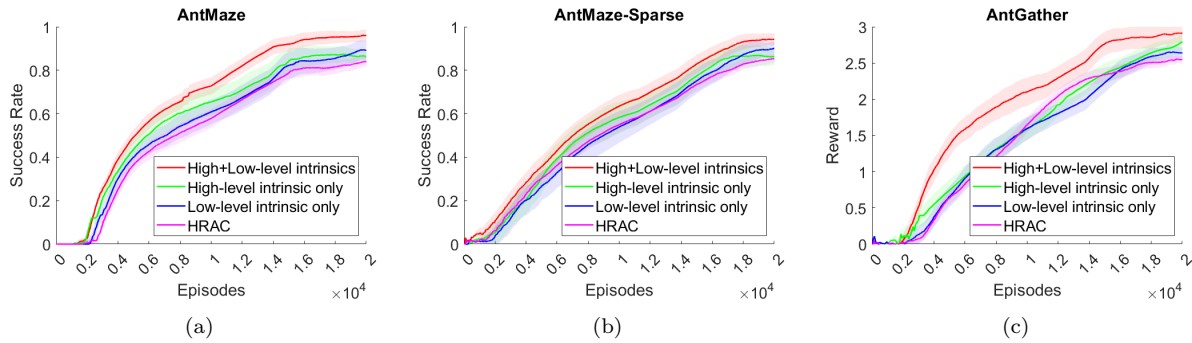

Figure 5: Success Rate on (a) AntMaze (b) AntMaze-Sparse and Reward on (c) AntGather using HRAC-G4RL, HRAC + High-level intrinsic, HRAC + Low-level intrinsic and HRAC. All curves have been smoothed equally for better visualization. The combination of high-level and low-level intrinsic rewards results in the highest success rates and fastest convergence.

Figures 4 to 7 show that, across all tested environments and algorithms, the combination of high-level and low-level intrinsic rewards results in the highest success rates and fastest convergence. The high-level intrinsic-only variant outperforms the low-level intrinsic-only variant, especially in sparse reward tasks, indicating that high-level intrinsic rewards play a crucial role in facilitating efficient exploration by encouraging the agent to select reachable and meaningful subgoals. In contrast, low-level intrinsic rewards have limited effect on exploration, primarily refining the execution of local behaviors. These results demonstrate that intrinsic rewards at different hierarchy levels serve complementary functions, and their combination yields superior performance.

### 4.3.2 Balancing between time and performance

To assess the trade-off between computational efficiency and performance, we evaluate two acceleration strategies mentioned in Section 3.5. First, we vary the sampling frequency of node features by testing intervals of 1, 5, and 10 steps in HLPS. As shown in Figure 8, increasing the sampling interval substantially reduces computation time, as it decreases the number of interactions with the graph, with only minor degradation in success rates across both AntMaze and AntPush tasks. Second, we vary the proportion of training data used for the graph encoder-decoder, testing 50%, 75%, and 100% subsets. Results in Figure 9

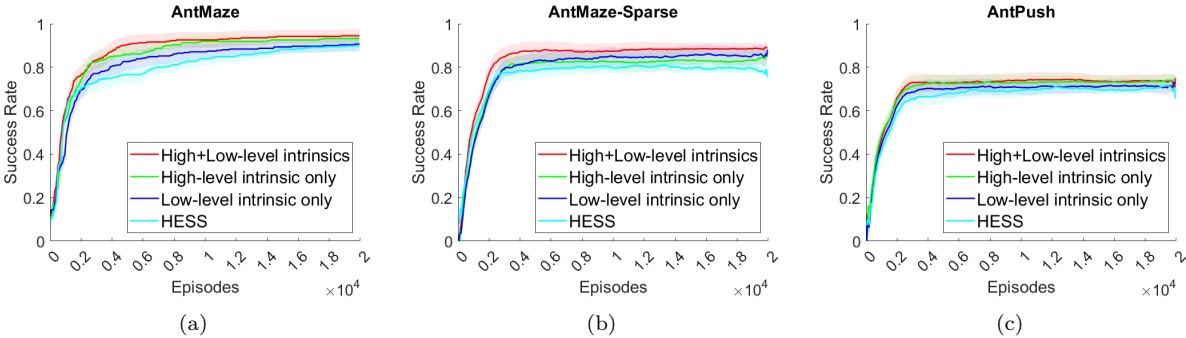

Figure 6: Success Rate on (a) AntMaze (b) AntMaze-Sparse and (c) AntPush using HESS-G4RL, HESS + High-level intrinsic, HESS + Low-level intrinsic and HESS. All curves have been equally smoothed for better visualization. The combination of high-level and low-level intrinsic rewards results in the highest success rates and fastest convergence.

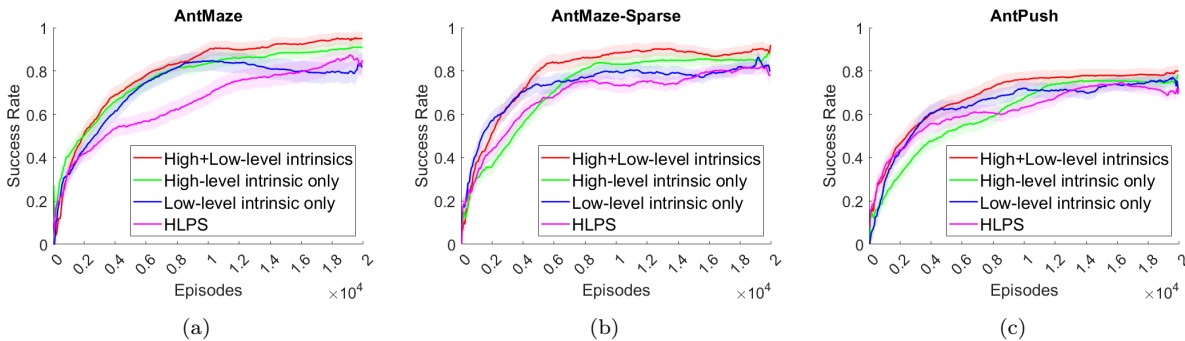

Figure 7: Success Rate on (a) AntMaze (b) AntMaze-Sparse and (c) AntPush using HLPS-G4RL, HLPS + High-level intrinsic, HLPS + Low-level intrinsic and HLPS. All curves have been equally smoothed for better visualization. The combination of high-level and low-level intrinsic rewards results in the highest success rates and fastest convergence.

indicate that reducing the amount of training data leads to only marginal improvements in computational efficiency and has negligible impact on final performance.

These findings suggest that the primary computational bottleneck of G4RL lies in the graph construction and node comparison processes described in Section 3.1.1, rather than in the encoder-decoder training itself. Adjusting the sampling frequency is therefore an effective approach for reducing time cost while largely preserving the benefits of G4RL integration.

## 4.4 Subgoal space visualization

This section shows how the subgoal space evolves in the AntMaze environment as the number of training episodes grows. We recorded state representations encountered in specific episodes and then used the corresponding graph encoders from those episodes to map these state representations to the subgoal representations. The subgoal representations are projected into 2D using t-SNE for visualization.

The distributions of subgoal representations in the subgoal space across different episodes are shown in Figure 10.

From the figure we can conclude that the graph encoder-decoder gives better subgoal representations when the number of episodes grows. At the 2000th episode, the subgoal representations form a few chains in the

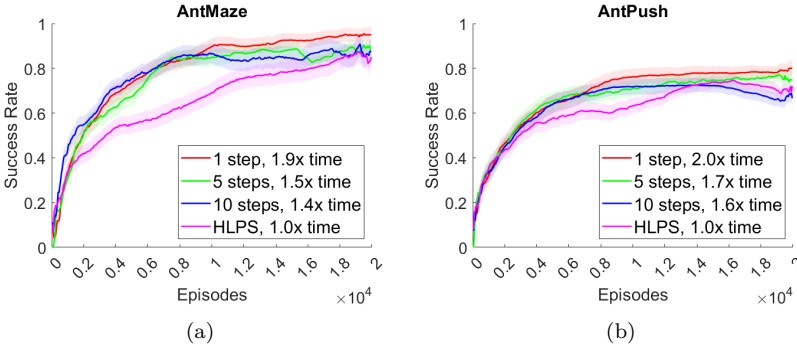

Figure 8: Success Rate on (a) AntMaze and (b) AntPush using HLPS+G4RL and HLPS. The number of steps in the legend indicates the selection of $t_c$ as described in Section 3.5 and the timescale is calculated w.r.t. the vanila HLPS algorithm. Increasing the sampling interval substantially reduces computation time, with only minor degradation in success rates across both AntMaze and AntPush tasks.

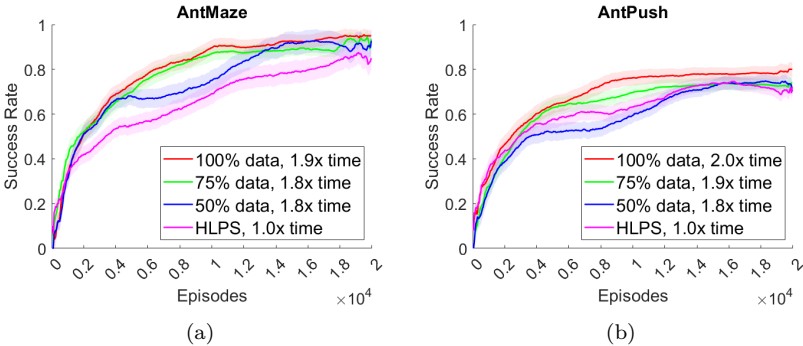

Figure 9: Success Rate on (a) AntMaze and (b) AntPush using HLPS+G4RL and HLPS. The percentage of data in the legend indicates the amount of data used in the training of the graph encoder-decoder as described in Section 3.5 and the timescale is calculated w.r.t. the vanila HLPS algorithm. Reducing the amount of training data leads to only marginal improvements in computational efficiency and has negligible impact on final performance.

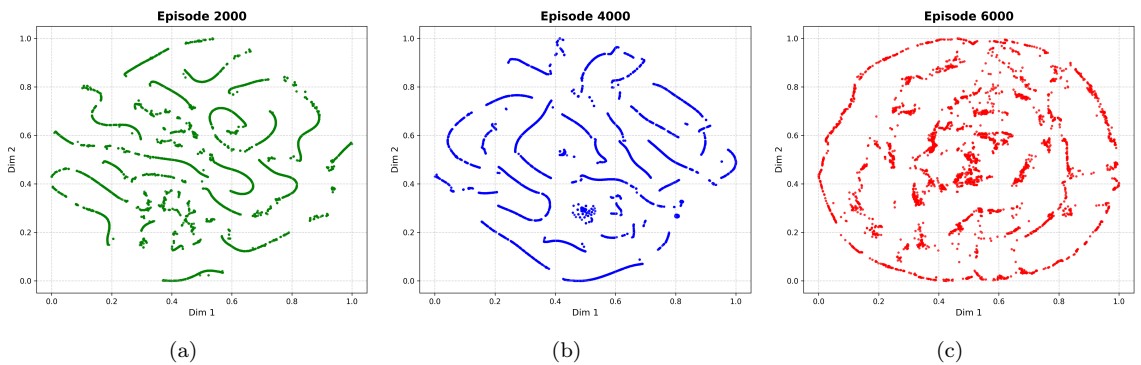

Figure 10: The distribution of subgoal representations in the subgoal space projected by t-SNE at the (a) 2000th (b) 4000th (c) 6000th episode. The subgoal space evolves as the training comes into later episodes.

2D space, indicating that the current graph is partitioned into many isolated sub-graphs. The graph encoder

has not yet learned to understand the environment well, and the agent cannot find relationships between different state representations yet. By the 4000th episode, the representations are less isolated, suggesting that the graph encoder has begun to learn the connections between different groups of state representation. Finally, at the 6000th episode, the subgoal representations are distributed more evenly across the entire space, indicating that the encoder can organize them into a highly connected graph. The subgoal space now better represents the underlying problem's structure.

The evolution of the graph encoder demonstrates that it can indeed learn to represent the task structure as the number of episodes grows.

## 5    Conclusion

We have presented a novel approach using a graph encoder-decoder to address the challenges of poor subgoal representations and sample inefficiency in GCHRL. The proposed architecture is designed to efficiently evaluate unseen states by operating in the graph representation space. It is easy to implement and can be seamlessly integrated into any existing GCHRL algorithms to enhance their performance in primarily symmetric environments. Our experiments on both sparse and dense control tasks have demonstrated the effectiveness and robustness of our method.

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

# A    Limitations and Future work

Despite the advantages demonstrated by G4RL in HRL tasks, its effectiveness depends significantly on several hyperparameters (e.g., $\epsilon_d$, $\alpha_l$, and $\alpha_h$), which require careful tuning to achieve optimal performance across different environments.

In future work, we aim to develop methods for automatically selecting these hyperparameters based on environmental dynamics, thereby reducing the need for manual tuning. Also, we plan to extend our work by exploring how to generate subgoals with more interpretable representations to facilitate knowledge transfer, potentially leveraging alternative graph representations (e.g. Graph Laplacian). Another promising direction is to transfer the knowledge embedded in the graph structure to new tasks by analyzing graph topology and establishing mappings between nodes of different state graphs. Additionally, more evidence on asymmetric environments is needed to demonstrate G4RL's robustness when asymmetric/irreversible transitions are present.

# B    Implementation details

## B.1    Environment details

**AntMaze** This environment is a part of the Gymnasium-Robotics libraries. The size of the environment is $24 \times 24$. Both of the state space and the action space are continuous, with a state dimension of 31 and an action dimension of 8. The reward of each step is defined by its negative Euclidean distance from the current location to the target position. At evaluation time, the goal is set to $(0, 16)$ and an episode is recognized as successful if the agent is within an Euclidean distance of 5 from the goal.

**AntMaze Sparse** This environment is a variant of Antmaze. The size of the environment is $20 \times 20$. The state and action spaces are the same as those in AntMaze. The reward of each step is 1 if and only if the agent reaches within an Euclidean distance of 1 from the goal, which is set to $(2, 9)$.

**AntGather** This environment is described in Duan et al. (2016). The size of the environment is $20 \times 20$. The state and action spaces are continuous. The task involves gathering apples to the designated place. The agent will be awarded $+1$ for each apple gathered and $-1$ for each bomb gathered. Apples and bombs are randomly placed in the $20 \times 20$ world.

**AntPush** The size of the environment is $20 \times 20$. The state and action spaces are continuous. A challenging task that requires both task and motion planning. The agent needs to move to the left then move up and push the block to the right in order to reach the goal.

## B.2    Network architecture details

Our network architecture for the HRL agents is the same as described in Nachum et al. (2018b), Zhang et al. (2022), Li et al. (2022) and Wang et al. (2024). For HIRO, HRAC, HESS and HLPS both the high-level and low-level agents use the TD3 algorithm. The size of each hidden layer in actor and critic networks in TD3 is 300.

For the graph encoder-decoder, we use a four-layer fully connected network with the hidden size of 128. The activation function used is ReLU. The decoder is a dot product of the two input subgoal representations.

We use Adam as the optimizer for the actor network, critic network and the graph encoder is Adam optimizer (Kingma & Ba, 2014).

## C   Hyperparameters

In this section we list all hyperparameters used in our experiments.

| Hyperparameters | Values |
|---|---|
| High-level agent | |
| Actor learning rate | 0.0001 |
| Critic learning rate | 0.001 |
| Batch size | 128 |
| Discount factor $\gamma$ | 0.99 |
| Policy update frequency | 1 |
| High-level action frequency | 10 |
| Replay buffer size | 20000 |
| Exploration strategy | Gaussian($\sigma = 1$) |
| Low-level agent | |
| Actor learning rate | 0.0001 |
| Critic learning rate | 0.001 |
| Batch size | 128 |
| Discount factor $\gamma$ | 0.99 |
| Policy update frequency | 1 |
| Replay buffer size | 20000 |
| Exploration strategy | Gaussian($\sigma = 1$) |

Table 1: Hyperparameters used in high- and low-level TD3 agents.

| Hyperparameters | Values |
|---|---|
| Number of nodes $N$ | 200 |
| Batch size | 128 |
| Optimizer learning rate | 0.0001 |
| $\epsilon_d$ | 0.1 for AntMaze/0.2 for others |
| $\alpha_h$ | 0.1 |
| $\alpha_l$ | 0.1 |
| $\beta$ | 0.2 |

Table 2: Hyperparameters used in the graph encoder-decoder.

## D    Algorithm

Now we describe our proposed method in Algorithm 1.

---
**Algorithm 1** GCHRL+G4RL
---
**Require:** High-level policy $\pi_h(g|\phi(s))$, low-level policy $\pi_l(a|\phi(s), g)$, replay buffer $\mathcal{B}$,
       graph encoder $\mathbf{E}$, graph decoder $\mathbf{D}$, high-level action frequency $K$,
       significance hyperparameter $\alpha_h$ and $\alpha_l$, tolerance hyperparameter $\beta$,
       number of episodes $N$, nunber of steps in one episode $T$.

1:  $n = 0$
2: **while** $n \leq N$ **do**
3:     $t = 0$
4:     $c = 0$
5:     **while** $t \leq T$ **do**
6:         **if** $t \mod K = 0$ **then**
7:             Execute the high-level policy $\pi_h(g_t|\phi(s_t))$ to sample the subgoal $g_t$.
8:         **else**
9:             Keep the subgoal $g_t$ unchanged.
10:        Execute the low-level policy $\pi_l(a_t|\phi(s_t), g_t)$ to sample the atomic action $a_t$.
11:        Sample reward $r_t$ and next state $s_{t+1}$.
12:        Calculate $r_h(s_t, g_t, s_{t+1})$ and $r_l(s_t, g_t, a_t, s_{t+1})$ using (9) and (10).
13:        Collect experience $(s_t, g_t, a_t, r_h, r_l, s_{t+1})$ and update the replay buffer $\mathcal{B}$.
14:        Update node representations and edge weights using collected experience.
15:        Update $c$ using (8).
16:        **if** $c \geq \beta$ **then**
17:            Update graph encoder $\mathbf{E}$ with node representation and edge information in the graph.
18:            $c = 0$.
19:        $t = t + 1$
20:     Update low-level policy $\pi_l(a|\phi(s), g)$ using the chosen HRL algorithm.
21:     Update high-level policy $\pi_h(g|\phi(s))$ using the chosen HRL algorithm.
22:     $n = n + 1$
---

## E Un-smoothed experiment results

In this section, we present the un-smoothed versions of figures 4 and 5 in the main text to show the full details of the experiments.

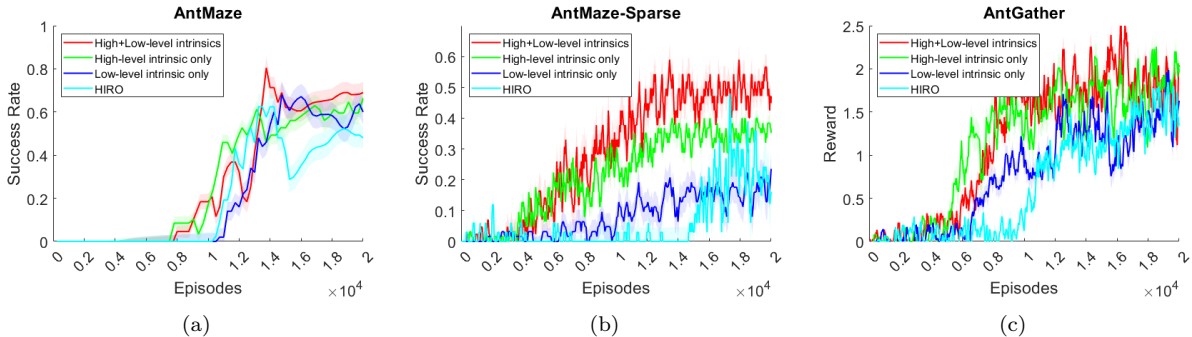

Figure 11: Success Rate on (a) AntMaze (b) AntMaze-Sparse and Reward on (c) AntGather using HIRO-G4RL, HIRO + High-level intrinsic, HIRO + Low-level intrinsic and HIRO. Smoothing is removed in contrast to figure 4

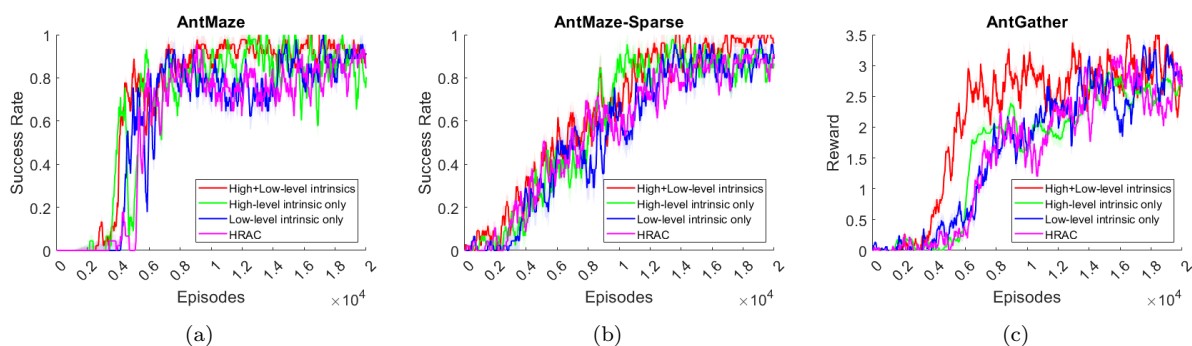

Figure 12: Success Rate on (a) AntMaze (b) AntMaze-Sparse and Reward on (c) AntGather using HRAC-G4RL, HRAC + High-level intrinsic, HRAC + Low-level intrinsic and HRAC. Smoothing is removed in contrast to figure 5

## F  Exploring the subgoal space through diverse visualization techniques

In this section, we first extend the visualization results presented in Figure 10 by applying multiple dimensionality reduction techniques—namely PCA, t-SNE, and UMAP—to demonstrate that the observations discussed in Section 4.4 are not specific to any single method. The results are shown in Figure 13 to 15.

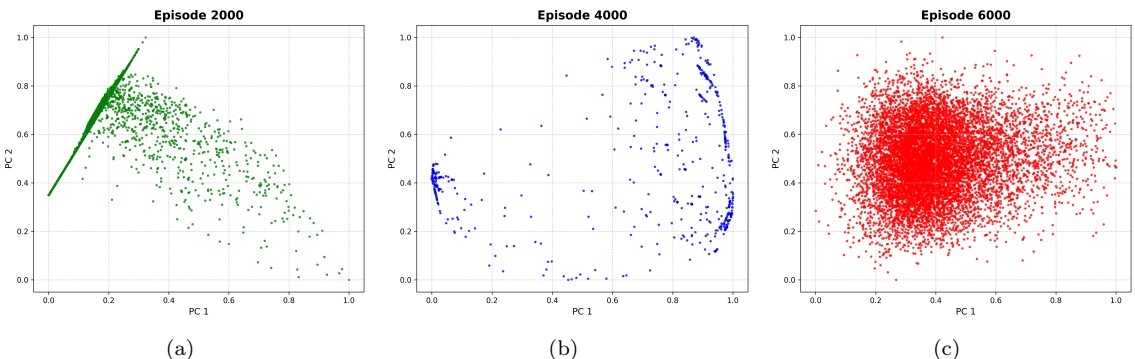

Figure 13: The distribution of subgoal representations in the subgoal space projected by PCA at the (a) 2000th (b) 4000th (c) 6000th episode.

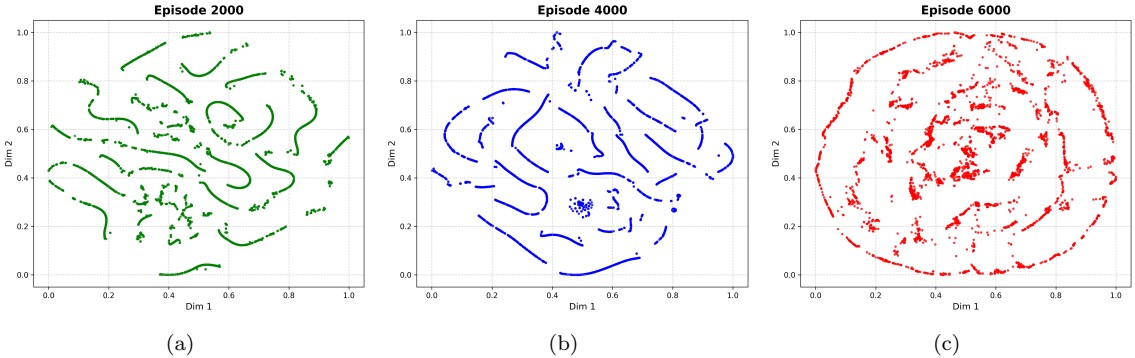

Figure 14: The distribution of subgoal representations in the subgoal space projected by t-SNE at the (a) 2000th (b) 4000th (c) 6000th episode.

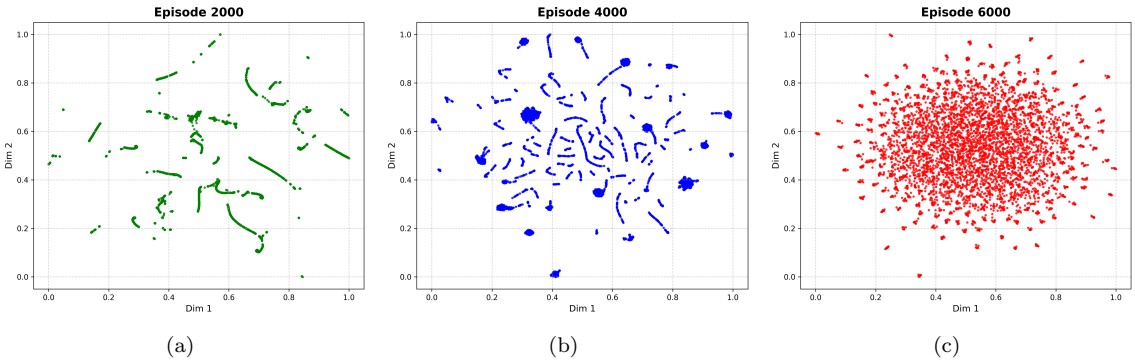

Figure 15: The distribution of subgoal representations in the subgoal space projected by UMAP at the (a) 2000th (b) 4000th (c) 6000th episode.

While the precise projections vary across methods, a consistent trend emerges: the representations progressively evolve from forming scattered clusters to a more distributed pattern. This trend aligns with the objective of G4RL—to learn a subgoal representation extractor that structures subgoals within a space functioning as a cognitive map, effectively capturing the underlying structure of the original state space. Such a map can be interpreted as an internal model of the environment, offering valuable information for planning and decision-making.

Figures 13 to 15 illustrate how the graph encoder progressively learns to utilize the subgoal space more effectively. Initially, it relies on only a limited region of the space, but over time, its usage becomes more distributed. This progression further supports the claim that the learned subgoal representations are meaningful and hold potential for providing continuous guidance.

We then visualize trajectories sampled at different stages of training in the AntMaze environment, projecting $\phi(s_t)$ into the state space and $g(s_t)$ into the subgoal space, respectively. Results are shown in Figures 16 to 19.

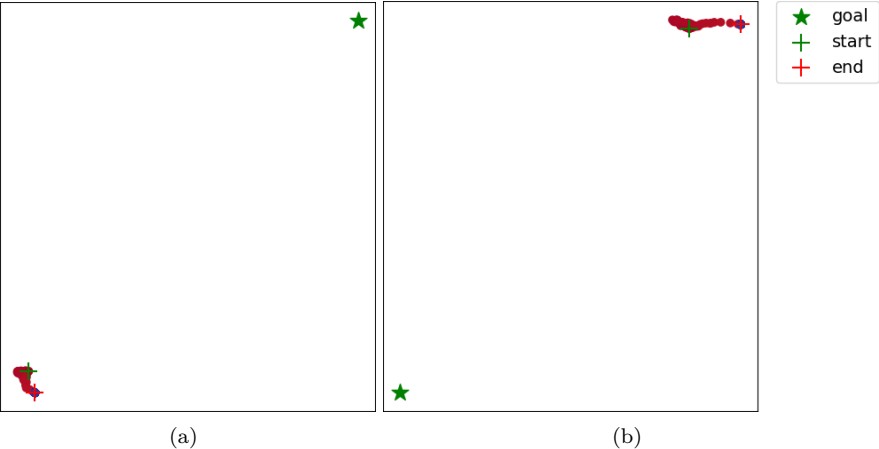

(a)                 (b)

Figure 16: Projection of (a) state representations $\phi(s_t)$ in the state space and (b) subgoal representations $g(s_t)$ in the subgoal space at the 2000th episode. The colour spectrum indicates the relative time step (red corresponds to earlier steps), while the green and red crosses mark the projected start and end points, respectively. The green star denotes the projected ultimate goal in the same space.

Figures 16 to 19 demonstrate that the learned subgoal space evolves progressively to reflect the agent's observations of the state space. A clear correspondence emerges between the connection structure of state representations $\phi(s_t)$ in the state space and subgoal representations $g(s_t)$ in the subgoal space. This supports the claim that the subgoal space captures the connectivity of the original state space and that the proposed subgoals offer accurate and consistent guidance toward the final goal.

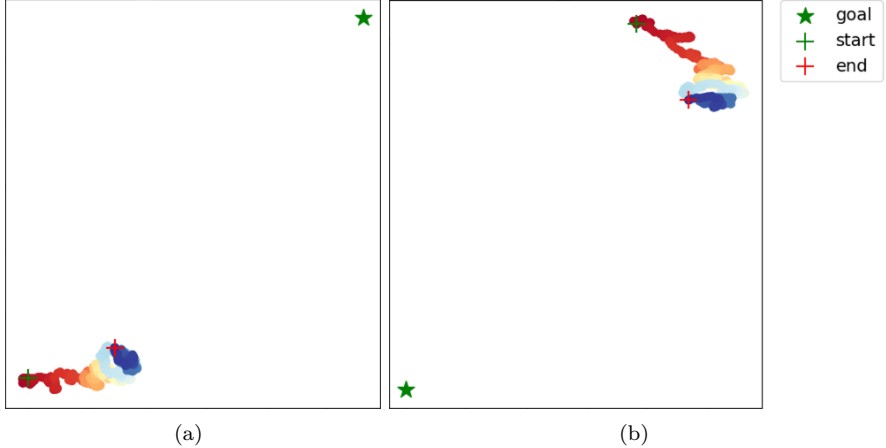

Figure 17: Projection of (a) state representations $\phi(s_t)$ in the state space and (b) subgoal representations $g(s_t)$ in the subgoal space at the 4000th episode. The colour spectrum indicates the relative time step (red corresponds to earlier steps), while the green and red crosses mark the projected start and end points, respectively. The green star denotes the projected ultimate goal in the same space.

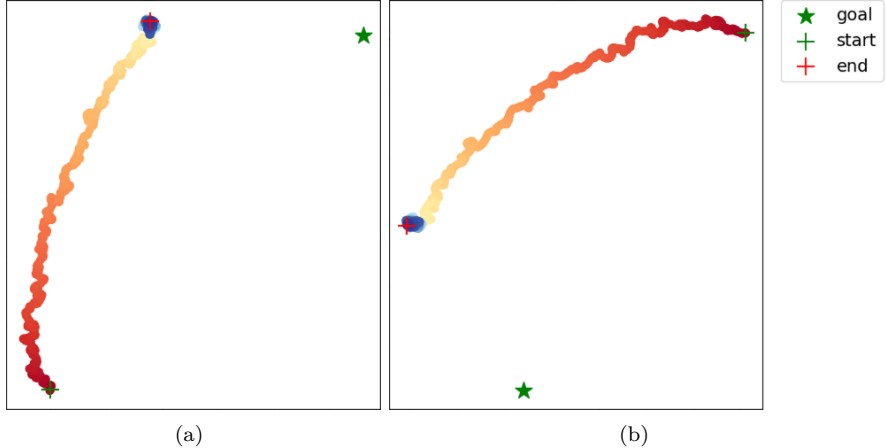

Figure 18: Projection of (a) state representations $\phi(s_t)$ in the state space and (b) subgoal representations $g(s_t)$ in the subgoal space at the 6000th episode. The colour spectrum indicates the relative time step (red corresponds to earlier steps), while the green and red crosses mark the projected start and end points, respectively. The green star denotes the projected ultimate goal in the same space.

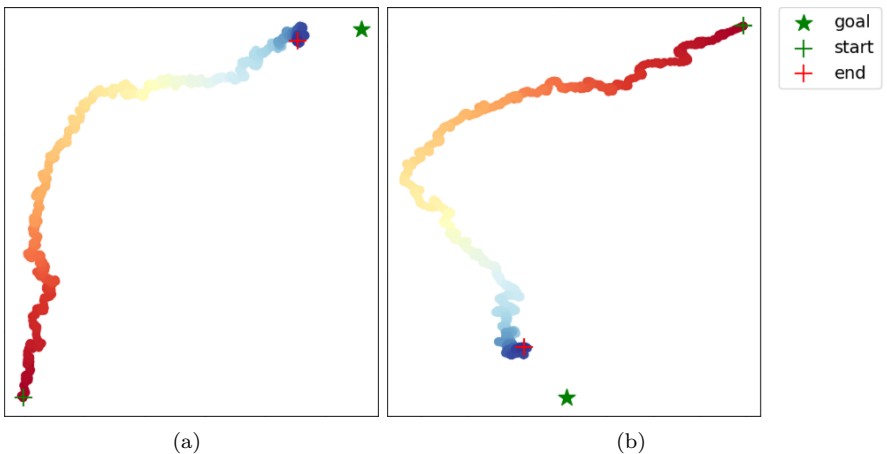

(a)                                                 (b)

Figure 19: Projection of (a) state representations $\phi(s_t)$ in the state space and (b) subgoal representations $g(s_t)$ in the subgoal space at the 8000th episode. The colour spectrum indicates the relative time step (red corresponds to earlier steps), while the green and red crosses mark the projected start and end points, respectively. The green star denotes the projected ultimate goal in the same space.

