# OpenReview forum: "Incorporating Spatial Information into Goal-Conditioned Hierarchical Reinforcement Learning via Graph Representations"
_TMLR — Accepted by TMLR_

### Review · Reviewer_v6Rq · 2025-05-28

**Summary Of Contributions:**

The paper addresses several issues with previous GCHRL methods, such as better utilization of graph representations and formulation of better graph-based sub-goal mechanisms. The proposed method is a graph autoencoder to represent sub-goals (thus allowing to encode “new unseen states”), from which they design a novelty-based intrinsic reward for improved exploration and sample-efficiency. Notably, the “decoder” is designed as the inner product between two state representations to predict their adjacency value, making it possible to predict how close two states are.

**Audience:**

Yes

**Claims And Evidence:**

No

**Requested Changes:**

* Overall, I’m not certain the paper provides enough evidence for its claims, for two main reasons: (1) the writing and language make it seem like a “project report” rather than a journal paper; and (2) the experimental results are limited to few state-based environments, and while it seems that they somewhat outperform previous methods, it comes at a computational cost (>=X 1.5). My main request here is to improve the writing and maybe present the results in a more convincing manner (e.g., by adding more discussion to the results sections as I mentioned above).
* See “Weaknesses” and “Minor” above.

**Strengths And Weaknesses:**

**Strengths**:
* Method is conceptually simple.
* Outperforms previous GCHRL methods (though sometimes it seems marginal).

**Weaknesses**:
* In general, the writing of this paper could benefit from extensive polishing (“a target state that the high-level agent wants the low-level agent to achieve as a part of the overall task.” For example, I don’t think “wants” is a formal language for a paper. There are many such examples throughout the paper.). It feels more like a “project report” rather than a paper.
* Method is simple in essence, but includes complex implementation details such as tuning the adaptive training schedule of the network. I think that Section 3.3 requires some more clarifications, the schedule design was not entirely clear to me.
* The method introduces increased time complexity due to training and updating the graph. While the authors propose a way to alleviate this by performing the operations every pre-defined time interval, I believe that with more complex tasks this will be detrimental to the performance.
* Simple and limited environments and tasks (mostly aligning with previous work). State-based only (i.e., no image-based inputs).
* I believe the results section requires more discussion.
* Lacking discussion of limitations.

**Minor**:
* Notations: Section 2.2 – I found the notations $g_t$ and $g(s_t)$ very confusing and I highly recommend choosing a different notation for $g(s_t)$.
* Page 4, first paragraph: I find the statement “This approach provides more accurate reward signals and leads to more reasonable target subgoals.” a bit too strong. If the author could provide more insights as to why, or better yet, a reference to empirical results that prove that, it would be great. Otherwise, I suggest toning it down.
* Page 4: “We perform the GCHRL explore process” -> “exploration process”
* Figure 1: something is wrong with the legend. While all graphs include all colors, the legend is divided between figures.
* Figure 7: why 5 steps take longer than 10 steps? Or is it a mistake in the figure?

---

> ### Author Response · Authors · 2025-06-12
> **Response to reviewer v6Rq (1/2)**
>
> We greatly appreciate your valuable comments and suggestions and have revised the manuscript accordingly.
>
> The responses to your comments are presented below.
>
> 1. "In general, the writing of this paper could benefit from extensive polishing."
>
> Response: We have thoroughly refined the paper to enhance its overall clarity, coherence, precision, flow and readability.
>
> 2. "Simple and limited environments and tasks (mostly aligning with previous work). State-based only (i.e., no image-based inputs)."
>
> Response: We have addressed this concern by adding experiments on image-based environments in Section 4.2. The results (see the new Figure 3) show that incorporating G4RL improves performance across all tested image-based environments.
>
> Regarding the choice of environments in our original experiments, we deliberately aligned with prior work to ensure a fair and consistent comparison between the backbone algorithms (HIRO, HRAC, HESS, and HLPS) and their G4RL-augmented versions. This allowed us to use the same environments, hyperparameters, and codebases provided by the original studies. Our goal was to demonstrate that G4RL can consistently enhance the performance of these backbone algorithms under comparable conditions.
>
> 3."Method is simple in essence, but includes complex implementation details such as tuning the adaptive training schedule of the network. I think that Section 3.3 requires some more clarification, the schedule design was not entirely clear to me."
>
> Response: We have substantially revised Section 3.3 to improve clarity, with particular focus on explaining the design of the adaptive training schedule. We believe the updated description provides a clearer understanding of the scheduling mechanism.
>
> 4."The method introduces increased time complexity due to training and updating the graph. While the authors propose a way to alleviate this by performing the operations every pre-defined time interval, I believe that with more complex tasks this will be detrimental to the performance."
>
> Response: We acknowledge your concern regarding the increased time complexity introduced by training and updating the graph. Our experimental results show that incorporating G4RL into the four GCHRL algorithms discussed in the paper increases the total runtime by a factor of approximately 1.9× to 2.5×,
> and by 1.2× to 1.6× when the proposed alleviation techniques are applied.
> This level of overhead is acceptable within the GCHRL domain. For instance, HLPS requires 4–5× more computation time than HESS, often for only marginal performance improvements.
>
> Moreover, since GCHRL algorithms have not yet been applied to more complex environments, it remains unclear how the integration of G4RL would affect time complexity in those settings. We have added a remark on this
> at the end of Section 4.3.2. Nonetheless, based on our current results, we suspect the added computational cost is unlikely to present a major obstacle.
>
>
> 5. "I believe the results section requires more discussion. Lacking discussion of limitations."
>
> Response: We have expanded the discussion in the results section and added a dedicated limitations paragraph in the final section of the paper (Section 5), as shown below:
>
> Despite the advantages demonstrated by G4RL in HRL tasks, its effectiveness depends significantly on several hyperparameters (e.g., $\epsilon_d$, $\alpha_l$, and $\alpha_h$), which require careful tuning to achieve optimal performance across different environments.
> In future work, we aim to develop methods for automatically selecting these hyperparameters based on environmental dynamics, thereby reducing the need for manual tuning.
>
> 6. "Figure 1: something is wrong with the legend. While all graphs include all colors, the legend is divided between figures."
>
> Response: To avoid overlapping between the curves and the legend box, we split the legend across subplots and mentioned this in the paper. However, we now realize that this approach caused confusion. In the revised version, we have updated Figure 1 by placing a unified legend box within one of the subplots to ensure clarity and avoid overlap.

---

> ### Author Response · Authors · 2025-06-12
> **Response to reviewer v6Rq (2/2)**
>
> 7. "Page 4, first paragraph: I find the statement “This approach provides more accurate reward signals and leads to more reasonable target subgoals.” a bit too strong. If the author could provide more insights as to why, or better yet, a reference to empirical results that prove that, it would be great. Otherwise, I suggest toning it down."
>
> Response: Our original statement was based on the improved performance observed when incorporating G4RL, as well as the qualitative insights provided in Figure 10 (Figure 9 in the original version).
> However, we acknowledge that the original phrasing may have overstated the conclusion without sufficient justification.
> In the revised version, we have modified the sentence to:
> "Additionally, we provide empirical evidence that the generated state graph accurately represents the underlying structure and relationships within the task environments," which is more accurate and better supported by the results shown in Figure 10.
>
>
> 8. Page 4: “We perform the GCHRL explore process” -> “exploration process”
>
> Response: Done.
>
>
> 9."Figure 7: why 5 steps take longer than 10 steps? Or is it a mistake in the figure?"
>
> Response: Updating the graph every 5 steps results in more frequent interactions with the graph compared to every 10 steps.. For example, over 200 environment steps,
> a 5-step interval would lead to 40 updates, whereas a 10-step interval would result in only 20. This increased frequency leads to higher computation time.
> We realize this may not have been clearly explained in the original version. To clarify, we have added the following sentence to the revised manuscript:
>  "increasing the sampling interval substantially reduces computation time, as it decreases the number of interactions with the graph".

---

> ### Author Response · Authors · 2025-08-01
> **Looking forward to feedbacks**
>
> Dear reviewer v6Rq,
>
> We would greatly appreciate any further comments you may have based on our responses and the revised version of the manuscript, and we look forward to your continued feedback.

---

> > ### Comment · Reviewer_v6Rq · 2025-08-18
> >
> > I thank the authors for their effort in revising the paper and clarifying my concerns. Overall, I think the paper reads much better now. I still feel like the experiments part is a bit weak and it remains an open question whether the additional complexity is justified for the reported performance; however, the contribution of incorporating graphs in GCHRL is clear and well-motivated.

---

> > > ### Author Response · Authors · 2025-08-20
> > >
> > > Dear reviewer v6Rq:
> > >
> > > Thank you for your invaluable feedback and thoughtful suggestions. We are pleased that our responses have addressed many of your concerns. Your constructive comments have significantly enhanced the quality of this paper, making it much stronger than the original version. We deeply appreciate the opportunity to refine our work and remain open to any further questions or suggestions you may have.
> > >
> > > Best regards,
> > > The Authors

---

### Review · Reviewer_ZrNt · 2025-07-15

**Summary Of Contributions:**

This paper provides a method for use in Goal-Conditioned Hierarchical Reinforcement Learning that tackles issues of high-level exploration,  effective subgoal space creation and sample efficiency (of both the high- and low-level agents). This method can be added to any existing method and uses a graph-based method for mapping spatial information to subgoal representations; a spatial graph that leverages the state representation and the connectivity to make the mapping to subgoals is maintained and updated (as new states are visited). Experiments were carried out on several variants of the Ant-X environments on MuJoCo, with baselines from various techniques; the experiments suggest that this additive method provides meaningful improvements.

**Audience:**

Yes

**Claims And Evidence:**

Yes

**Requested Changes:**

- Page 1: There seems to be a link attached to the footer / page number on Page 1; is this intentional?
- Introduction, Paragraph 1; "its resemblance to the human thinking process" -> Is there a reference for this statement?
- Introduction, Paragraph 3; " no prior work has attempted to address them in a unified framework." -> This sentence suggests that your method is a 'unifying framework' — is this true? If so, are you trying to unify the:
	1.  "enhance(ment of) the efficiency of high-level exploration"
	2. "find(ing of) a good subgoal space to boost GCHRL’s performance in complex environments"
	if not, then please be more specific.
- Introduction, Paragraph 3; "Yet, such integration is crucial, as uniting an effective high-level agent with an accurate low-level agent within a cohesive framework can yield performance improvements that surpass simple additive effects." -> This sentence makes it sound like other works don't train the low-level and high-level agent jointly and cohesively. Is this true to say?
- Introduction, Paragraph 5; "Additionally, most existing works..." -> Please add references to "most existing works...".
- Section 2.1; "Notation" -> Perhaps a better title for this section is "Reinforcement Learning" or "Markov Decision Processes".
- Section 2.2, Paragraph 1; "Goal-Conditioned Reinforcement Learning (GCRL) trains agents to achieve specific goals, which are the target states. The agent receives an additional goal input along with the state input and learns a policy that aims to achieve this goal. Goals are represented explicitly in the input to the policy, guiding the agent’s actions towards desired outcomes" -> Wouldn't it make sense to add math notation here? E.g., "The aganet receives an additional goal input $g_t$ along with the state input $s_t$ and learns a policy $\pi(a_t | g_t, s_t)$ that aims to achieve this...".
- Section 2.2, Paragraph 2 -> Is the norm missing a subscript 2?
- Equation (2) -> According to this equation, is the graph stored in terms of the raw state $s_t$, or the state representation $\phi(s_t)$? If not the state representation, what is the reason? Since the state representation is used in computing the distances in Equation (1) and Equation (3), wouldn't it be more computationally efficient to store the state representation?
- Figure 1 -> can the legend be moved to the top of the figures.
- Figure 2, 3 -> can the arrangement of the legends be the same as Figure 1.
- Figure 4 and 5 -> Please add the fact that the curves were smoothed in the caption. Furthermore can the un-smoothed figures be added to an Appendix for verification that any conclusions weren't affected by the smoothing.
- Figures in general -> I think it is useful to write the main conclusion of each figure in the figure caption.

**Strengths And Weaknesses:**

Please note that I am not confident in commenting on the novelty / strength of the contribution of this work.
### Strengths
- Good Preliminaries section.
- I think Section 3 is really written well with each major component of the proposed method being given text-space to be explained clearly.
- Appreciate Section 3.5 that analyses the speed / performance trade off in their method.
- Extremely pleased to see that a large number of repetitions of the experiments was carried out!
- Good experiments with good ablation studies.

#### Weaknesses
- The main weakness is in writing style and consistency of the exposition. These can be easily fixed — see Requested Changes below.
- Are you using a directed or undirected graph? From Equation (4) I expect it is an undirected graph? If so, why is this a better choice than a directed graph; I would expect that state transitions aren't generally symmetric.
- From what I can see, code isn't made available.

---

> ### Author Response · Authors · 2025-07-17
> **Response to reviewer ZrNt (1/2)**
>
> We greatly appreciate your valuable comments and suggestions and have revised the manuscript accordingly.
>
> The responses to your comments are presented below.
>
> **1. "Page 1: There seems to be a link attached to the footer / page number on Page 1; is this intentional?"**
>
> Response: This is not intentional. It seems that the template created the link, and we have fixed it.
>
> **2."Introduction, Paragraph 1; "its resemblance to the human thinking process" - Is there a reference for this statement?"**
>
> Response: We have added the following two references:
>
>
> Chi-Yang Hsu, Kyle Cox, Jiawei Xu, Zhen Tan, Tianhua Zhai, Mengzhou Hu, Dexter Pratt, Tianlong
> Chen, Ziniu Hu, and Ying Ding. Thought graph: Generating thought process for biological reasoning. In
> Companion Proceedings of the ACM Web Conference 2024, pp. 537–540, 2024.
>
> Hong Zhao, Yao Fu, Weihao Jiang, Shiliang Pu, and Xiaoyu Cai. Simulate human thinking: Cognitive
> knowledge graph reasoning for complex question answering. In Pacific-Asia Conference on Knowledge
> Discovery and Data Mining, pp. 522–534. Springer, 2022.
>
>
> **3. "Introduction, Paragraph 3; 'no prior work has attempted to address them in a unified framework.' - This sentence suggests that your method is a 'unifying framework' — is this true? If so, are you trying to unify the:
> ‘enhance(ment of) the efficiency of high-level exploration’
> ‘find(ing of) a good subgoal space to boost GCHRL’s performance in complex environments’ if not, then please be more specific."**
>
> Response: Yes, our method is indeed intended as a unifying framework. More specifically, we aim to jointly address (i) the enhancement of the efficiency of high-level exploration and (ii) the enhancement of the accuracy of the low-level learning signal. We have revised the sentence to clarify this point.
>
> **4. "Introduction, Paragraph 3; "Yet, such integration is crucial, as uniting an effective high-level agent with an accurate low-level agent within a cohesive framework can yield performance improvements that surpass simple additive effects." - This sentence makes it sound like other works don't train the low-level and high-level agent jointly and cohesively. Is this true to say?"**
>
> Response: You're correct — it would not be accurate to claim that prior works do not train high-level and low-level agents jointly. Our intention in this sentence was not to suggest that, but rather to emphasize that prior works generally do not aim to simultaneously enhance both the effectiveness of the high-level agent and the accuracy of the low-level learning signal within a unified framework. To clarify this, we have revised the sentence to: "as increasing the effectiveness of the high-level agent and the accuracy of the low-level agent within a cohesive framework can yield performance improvements that surpass simple additive effects."
>
> **5. "Introduction, Paragraph 5; "Additionally, most existing works..." - Please add references to "most existing works..."."**
>
> Response: We have added the following references:
>
> Zhang-Wei Hong, Tao Chen, Yen-Chen Lin, Joni Pajarinen, and Pulkit Agrawal. Topological experience
> replay. arXiv preprint arXiv:2203.15845, 2022.
>
> Deyao Zhu, Li Erran Li, and Mohamed Elhoseiny. Value memory graph: A graph-structured world model
> for offline reinforcement learning. arXiv preprint arXiv:2206.04384, 2022.
>
> **6. "Section 2.1; "Notation" - Perhaps a better title for this section is "Reinforcement Learning" or "Markov Decision Processes"."**
>
> Response: We have changed the title to "Markov Decision Processes".
>
> **7. "Section 2.2, Paragraph 1; "Goal-Conditioned Reinforcement Learning (GCRL) trains agents to achieve specific goals, which are the target states. The agent receives an additional goal input along with the state input and learns a policy that aims to achieve this goal. Goals are represented explicitly in the input to the policy, guiding the agent’s actions towards desired outcomes" - Wouldn't it make sense to add math notation here? "**
>
> Response: We have added mathematical notation as suggested to improve clarity.
>
> **8."Section 2.2, Paragraph 2 - Is the norm missing a subscript 2?"**
>
> Response: Yes, you are right. We have fixed this.
>
> **9. "Equation (2) - According to this equation, is the graph stored in terms of the raw state, or the state representation? If not the state representation, what is the reason? Since the state representation is used in computing the distances in Equation (1) and Equation (3), wouldn't it be more computationally efficient to store the state representation?"**
>
> Response: The graph is stored in terms of the state representation, and indeed, it is more computationally efficient to store the state representation.
> Once the environment gives a raw state, we transform it into a representation immediately and discard the raw state.

---

> > ### Comment · Reviewer_ZrNt · 2025-07-17
> > **Response to comments**
> >
> > Dear Authors,
> >
> > Thank you very much for addressing pretty much all of my comments, and so quickly!
> >
> > I had a couple more comments based on yours:
> >
> > 9. If the graph is storing the state representation, do you think that the notation $\textbf{A}_{\phi (s_t), \phi(s\_{t-1})}$ would make this clearer?

---

> > > ### Author Response · Authors · 2025-07-17
> > > **Response to additional comment**
> > >
> > > Dear reviewer ZrNt,
> > >
> > > Thank you greatly for your informative and timely reply!
> > >
> > > We have updated the notation in Equations (2), (4), and (7) as well as in the main text to emphasize that the node features are state representations rather than raw states.

---

> > > > ### Comment · Reviewer_ZrNt · 2025-07-17
> > > > **Responses to final comments**
> > > >
> > > > Hello!
> > > >
> > > > Perfect. Thank you very much!

---

> ### Author Response · Authors · 2025-07-17
> **Response to reviewer ZrNt (2/2)**
>
> **10. "Figure 1,2,3 - can the legend be moved to the top of the figures."**
>
> Response: We explored several legend placements during figure development, and after careful consideration of the impact, as well as feedback from another reviewer, we found that the current layout offers the best balance between clarity and consistency with the overall figure and paper format.
> Therefore, we have decided to retain the existing legend positions.
>
> **11. "Figure 2, 3 - can the arrangement of the legends be the same as Figure 1"**
>
> Response: Following your suggestion, we have removed the legends in the first two subplots of both Figures 2 and 3,
> to make them consistent with Figure 1.
>
> **12. "Figure 4 and 5 - Please add the fact that the curves were smoothed in the caption. Furthermore can the un-smoothed figures be added to an Appendix for verification that any conclusions weren't affected by the smoothing."**
>
> Response: We have added the note "All curves have been equally smoothed for better visualization." to the captions of Figures 4 and 5. Additionally, we have added the un-smoothed versions in Appendix D to confirm that the conclusions are unaffected by the smoothing.
>
> **13. "Figures in general - I think it is useful to write the main conclusion of each figure in the figure caption."**
>
> Response: We have added the main conclusion of each figure in the figure caption.
>
> **14. "Are you using a directed or undirected graph? From Equation (4) I expect it is an undirected graph? If so, why is this a better choice than a directed graph; I would expect that state transitions aren't generally symmetric."**
>
> Response: We are using an undirected graph, primarily for two reasons:
>
> Efficiency: An undirected graph requires fewer resources for storage and computation compared to a directed graph.
>
> Method compatibility: Our method relies on defining a deterministic distance between state/subgoal representations for each node pair. This is straightforward in an undirected graph but becomes problematic in a directed setting, where distances can be asymmetric or undefined. Such ambiguity hinders the graph encoder-decoder’s ability to converge, as it depends on a consistent target for each node pair.
>
> While it is possible to design a specialized encoder-decoder architecture that handles directed graphs, doing so introduces additional complexity and uncertainty. At this stage, we believe that the bias introduced by using an undirected graph is preferable to the potential instability caused by a more complex, directed alternative.

---

> > ### Comment · Reviewer_ZrNt · 2025-07-17
> > **Response to comments 2**
> >
> > 14. Would you be able to add to the text somewhere that you are using an undirected graph (I don't think I saw this in the updated text), as well as your justification?

---

> > > ### Author Response · Authors · 2025-07-17
> > > **Response to additional comment 2**
> > >
> > > We have added the third paragraph in Section 3.1:
> > >
> > > We choose the state graph to be undirected for mainly two reasons: **(1) Efficiency:** An undirected graph requires fewer resources for storage and computation compared to a directed graph, and **(2) Method compatibility:** Our method relies on defining a deterministic distance between state/subgoal representations for each node pair. This is straightforward in an undirected graph but becomes problematic in a directed setting, where distances can be asymmetric or undefined. Related details will be explained in Sections 3.2 and 3.3
> > >
> > > To show the fact and the reason why we are using an undirected graph.

---

### Review · Reviewer_aBsC · 2025-08-17

**Summary Of Contributions:**

The paper addresses the challenge of learning effective subgoal representations and improving exploration efficiency in Goal-Conditioned Hierarchical Reinforcement Learning (GCHRL). The key contributions can be summarized as follows:

1. Graph-Guided sub-Goal representation Generation (G4RL):
   The authors propose a novel architecture that integrates a graph encoder–decoder into GCHRL. The encoder maps state representations into a subgoal embedding space, while the decoder—implemented via dot product—ensures that the learned embeddings preserve adjacency relations in the state graph. This design enables the system to evaluate both visited and unseen states more effectively.

2. Online construction of state graphs:
   The method maintains an evolving undirected state graph during exploration, where nodes correspond to visited states and edges encode observed transitions. The graph is updated dynamically as training proceeds, avoiding reliance on handcrafted or pre-defined graphs.

3. Intrinsic rewards at both hierarchy levels:
   By leveraging distances in the learned subgoal space, the framework introduces novelty-based intrinsic rewards for both the high-level and low-level agents. This aims to guide the high-level agent toward reachable and meaningful subgoals and provide the low-level agent with more accurate progress signals.

4. Empirical validation across multiple benchmarks:
   Experiments are conducted on several MuJoCo tasks (AntMaze, AntGather, AntPush, AntFall, and AntMaze-Sparse) under both dense and sparse reward settings, as well as image-based state inputs. Results indicate that incorporating G4RL consistently improves sample efficiency and final success rates across multiple backbone GCHRL algorithms (HIRO, HRAC, HESS, HLPS).

5. Additional analyses:
   The paper includes ablation studies that separately examine the effects of high- vs. low-level intrinsic rewards, as well as computational trade-offs such as graph update frequency and training data subsampling. A visualization of the subgoal embedding space further illustrates how the learned representations evolve over training.

Overall, the submission contributes a general mechanism for embedding spatial information into subgoal representations via graph learning, which can be combined with existing GCHRL methods to enhance performance in long-horizon, sparse-reward environments.

**Audience:**

Yes

**Claims And Evidence:**

Yes

**Requested Changes:**

### Critical

* Clarify and justify the decoder design and undirected graph assumption:
  The paper currently employs a dot-product decoder with an undirected graph, and explains this mainly in terms of computational efficiency and compatibility. However, this design implicitly imposes strong assumptions on the environment’s dynamics:

  * It assumes that the adjacency matrix can be embedded into a symmetric Euclidean space (i.e., low-rank approximation).
  * It enforces undirected connectivity, thereby biasing the representation toward environments with reversible dynamics (e.g., physical navigation tasks), while making it unsuitable for settings with inherently asymmetric transitions (e.g., teleportation, one-way actions, irreversible state changes).

  This implicit bias directly affects the claim of being “applicable to any GCHRL method.” The authors should explicitly discuss these assumptions in the paper, clarify the extent to which their method is restricted to approximately symmetric/Euclidean domains, and justify why dot-product similarity is chosen over other possible decoders. Without such discussion, the generality claim remains questionable.


### Optional (would strengthen the work, but not critical for acceptance)

* Strengthen baselines:
  Since TD3 predictably fails in sparse-reward long-horizon tasks, incorporating stronger goal-conditioned baselines such as TD3+HER would provide a more balanced comparison.

* Broaden evaluation domains:
  The evaluation is limited to MuJoCo Ant-based tasks that share similar geometric properties. Adding results on domains with less geometric regularity (e.g., discrete rooms, symbolic or video game environments) would better support the claim of broad applicability.

**Strengths And Weaknesses:**

### Strengths

* Novel integration of graph learning with hierarchical RL:
  The paper introduces a graph encoder–decoder that enables subgoal representations to capture spatial connectivity among states. This provides a new perspective on addressing both high-level exploration inefficiency and low-level reward signal quality within a unified framework.

* Clear formulation and implementation details:
  The method is described systematically, including graph construction, updating, encoder–decoder training, and reward shaping. Pseudocode and hyperparameter tables in the appendix improve clarity and reproducibility.

* Extension to image-based inputs:
  Demonstrating applicability beyond low-dimensional state features adds credibility to the method within robotics-style environments.


### Weaknesses

* Decoder assumptions and symmetry restriction:
  The dot-product decoder implicitly assumes that the adjacency structure of the state graph can be embedded into a symmetric Euclidean space (i.e., low-rank approximation). This design enforces undirected graphs, which may be reasonable for reversible navigation tasks but limits applicability to environments with asymmetric or non-Euclidean connectivity (e.g., discrete rooms, teleportation, one-way transitions). As a result, the claim of being “applicable to any GCHRL method” may be overstated.

* Baseline limitations:
  The non-hierarchical baseline (TD3) predictably underperforms in long-horizon sparse-reward tasks. Without including stronger goal-conditioned baselines such as TD3+HER, it is difficult to fully assess the comparative advantage of G4RL.

* Evaluation domain narrowness:
  All experiments are conducted in MuJoCo Ant-based environments, which share similar geometric properties. This may bias results toward methods leveraging spatial embeddings, and leaves applicability to broader domains (e.g., strategic video games, non-robotics tasks) uncertain.

---

> ### Author Response · Authors · 2025-08-18
> **Response to reviewer aBsC (1/2)**
>
> We sincerely appreciate your valuable comments and suggestions, which have significantly helped to improve the quality of our paper. We have revised the manuscript accordingly, and below, we address each of your concerns:
>
> **1. "The claim of being “applicable to any GCHRL method” may be overstated"**
>
> Response: We appreciate the reviewer’s feedback and have revised the original claim to more accurately reflect the scope of our work. Specifically, we now state: "This architecture can be integrated into any GCHRL algorithm operating in environments with primarily symmetric and reversible transitions, enhancing performance across this class of problems."
>
> This updated phrasing appears in the abstract, Section 1 and the conclusion, and more accurately conveys the intended applicability based on the specific settings evaluated in our study.
>
> **2. "The authors should explicitly discuss these assumptions in the paper, clarify the extent to which their method is restricted to approximately symmetric/Euclidean domains."**
>
> Response: Thank you for your valuable feedback. We have included
> the following statements in Section 3.1 to clarify this issue:
> "This choice is based on the assumption that G4RL is designed for environments with symmetrical and reversible dynamics. However, some of our experiments demonstrate that G4RL can still enhance performance in partially asymmetric environments; please refer to the experiment section for further details."
>
> In the AntPush environment, for example, a box cannot be returned to its original position once it is pushed into a corner. Similarly, in AntFall, there is an elevated platform that the agent cannot access after falling off. These scenarios introduce asymmetric transitions, for instance, the transition between a state where the box is in a pushable position and a state where it is pushed into a corner in AntPush, or the transition from above to below the platform in AntFall. Despite this, our results in Figures 2,3,6,7,8 and 9
> show that G4RL still perform well in both environments.
> We have modified section 4 to explicitly talk about this issue.
>
>
> We acknowledge that these results may not fully demonstrate G4RL’s robustness in settings with inherently asymmetric or irreversible transitions. To address this, we have added the following statement in the "Future Work" section: "Additionally, further exploration of G4RL’s performance in environments with asymmetric or irreversible transitions is necessary to better assess its robustness in such settings."

---

> ### Author Response · Authors · 2025-08-18
> **Response to reviewer aBsC (2/2)**
>
> **3. "The authors should justify why dot-product similarity is chosen over other possible decoders"**
>
> Response: We chose dot-product similarity based on the assumption that the similarity between two nodes, such as the overlap in their local neighbourhoods, is well captured by the dot product of their embeddings.
> This assumption is supported by prior work in the graph embedding literature [1,2,3]:
>
>
> We have added this justification in Section 3.2 after Equation 6.
>
> In comparison to other common decoders, the dot-product approach offers several advantages:
>
> Neural Network (NN) Decoder: A neural network-based decoder requires additional training, introducing extra complexity and variability. Since the decoder’s parameters are updated during training, this can lead to instability in the learned representations and pose challenges in parameter optimization due to high variance [4].
>
> L2 Distance Decoder: The L2 Distance decoder is  $D(g(s_u),g(s_v)) = \|g(s_u)-g(s_v)\|^2_2$  and the corresponding loss term is $ \sum_{\phi(s_u),\phi(s_v) \in \mathcal{V}}(\mathbf{D}(g(s_u),g(s_v))*A_{\phi(s_u),\phi(s_v)}) $ . As the equation suggests, an L2 distance decoder cannot impose specific constraints on unconnected nodes, as the term will always be 0 for unconnected nodes in the loss function, which may lead to high variance in the learned representations for nodes with no connectivity to each other. In contrast, the dot-product decoder enforces a stronger constraint, requiring the representations of unconnected nodes to be orthogonal in the Euclidean space, which helps ensure more consistent and meaningful embeddings.
>
> In conclusion, a dot product decoder helps to produce more stable (compared to NN decoders) and meaningful (compared to L2 distance decoders) node representations, which reduces the variance and helps the encoder-decoder converge faster.
>
>
> Thank you again for your thoughtful comments. We hope that our responses have addressed your concerns, and we look forward to the possibility of a revised evaluation based on the clarifications provided.
>
> - [1] Ahmed, Amr, et al. "Distributed large-scale natural graph factorization." Proceedings of the 22nd international conference on World Wide Web. 2013.
>
> - [2] Cao, Shaosheng, Wei Lu, and Qiongkai Xu. "Grarep: Learning graph representations with global structural information." Proceedings of the 24th ACM international on conference on information and knowledge management. 2015.
>
> - [3] Ou, Mingdong, et al. "Asymmetric transitivity preserving graph embedding." Proceedings of the 22nd ACM SIGKDD international conference on Knowledge discovery and data mining. 2016.
>
> - [4] Zheng, Stephan, et al. "Improving the robustness of deep neural networks via stability training." Proceedings of the ieee conference on computer vision and pattern recognition. 2016.

---

> ### Comment · Reviewer_aBsC · 2025-08-19
>
> Thanks for the revision. It is indeed important for readers to understand why the dot-product decoder design works and its intuition for goal-conditioned RL.
>
> Regarding the subgoal visualization, you currently use PCA. I wonder whether alternative visualization techniques such as t-SNE or UMAP would reveal a similar structure. This would help verify whether the observed organization of subgoals is robust across projection methods, and also clarify what the expected structure of the encoder–decoder design should look like.
>
> Relatedly, does the design implicitly encourage a self-organizing map–like organization of subgoals (not necessarily in the strict SOM sense, but in the sense of forming topologically coherent clusters)? A short discussion in the main paper or appendix would make these assumptions clearer and more interpretable.

---

> > ### Author Response · Authors · 2025-08-21
> >
> > Dear Reviewer aBsC,
> >
> > Thank you for your insightful suggestions—they indeed help strengthen our work.
> >
> > We have revised Section 4.4 and added visualizations using PCA, t-SNE, and UMAP in Appendix F. These additional visualizations consistently show that the representations evolve from forming scattered clusters to a more evenly distributed pattern across the subgoal space.
> >
> > Regarding the idea of a self-organizing map, we agree that the subgoal space is intended to function as a cognitive map to support planning and decision-making in the original state space. We have added a brief discussion on this point in Appendix F. However, performing an explicit coherence analysis between the subgoal space and the original state space remains challenging, as the subgoal representation also encodes non-spatial information—such as joint angles and velocity in the case of the Ant robot.

---

> > ### Author Response · Authors · 2025-08-22
> >
> > Dear reviewer aBsC:
> >
> > We greatly appreciate your time and insights. Should you have any further feedback or concerns, we are more than happy to address them promptly. Please feel free to reach out if there is anything else you would like to discuss.
> >
> > Best regards,
> > The Authors

---

> > > ### Comment · Reviewer_aBsC · 2025-08-22
> > >
> > > I found that the revised Section 4.4 claims to use t-SNE for visualization, while the previous version mentioned PCA. Was that a typo in the earlier version?

---

> > > > ### Author Response · Authors · 2025-08-22
> > > >
> > > > Yes, it was indeed a typo in the original version — the figures were generated using t-SNE, but we mistakenly wrote PCA in the text. We apologize for the confusion.

---

> > > ### Comment · Reviewer_aBsC · 2025-08-23
> > >
> > > Thanks for the further explanation. If you want to go further, could you clarify the intent of the  plots—are they primarily meant to demonstrate connectivity (and thus suggest continuous guidance in the subgoal space)?
> > >
> > > To make the guidance effect more explicit, I suggest adding a per-episode path overlay that shows, for the same episode, the trajectories of $\phi(s_t)$ and $g(s_t)$ as two polylines in the same 2D projection (if they overlap too much, please show them in two separate panels). This simple side-by-side (or overlaid) path view would allow readers to directly see how subgoals guide state evolution over time. Any standard projection (PCA, t-SNE, or UMAP) would be suitable for this purpose.

---

> > > > ### Author Response · Authors · 2025-08-24
> > > >
> > > > Thank you for the suggestion. The subgoal projection plots are intended to illustrate how the graph encoder progressively learns to utilize the subgoal space more effectively. Initially, it focuses on a limited region, but over time, its usage becomes more distributed. This progression supports our claim that the learned subgoal representations are meaningful and have the potential to provide continuous guidance.
> > > >
> > > > Additionally, we have included new visualizations that project $\phi(s_t)$ and $g(s_t)$ from a single trajectory into the state space and subgoal space, respectively. These figures reveal a clear correspondence in connectivity and relative positioning between the two spaces, offering more direct evidence that the learned subgoal representations can provide accurate and continuous guidance to the agent.
> > > >
> > > > Please let us know if you have any further questions or suggestions.

---

> > > > > ### Comment · Reviewer_aBsC · 2025-08-25
> > > > >
> > > > > Thanks for your revision. The new visualizations provide strong evidence that the proposed method indeed achieves its intended effect, beyond just performance improvement. I have no further suggestions, but if you would like to slightly polish the new trajectory plot, using consistent coordinate alignment could help interpretability, since  and  share the same feature space for the low-level policy objective.

---

> > > > > > ### Comment · Reviewer_aBsC · 2025-08-25
> > > > > >
> > > > > > fix my comment typo: \* since $\phi(s)$ and $g(s)$ share the same feature space for the low-level policy objective.

---

> > > > > > ### Author Response · Authors · 2025-08-25
> > > > > >
> > > > > > Thank you for your thoughtful and forward-looking suggestions — they served as valuable "subgoals" and significantly improved the quality of our paper compared to the initial submission.
> > > > > >
> > > > > > We sincerely appreciate the time and effort you dedicated to our work. The review process was both insightful and genuinely enjoyable!

---

### Decision · Action_Editor_rdGn · 2025-09-22

**Recommendation:** Accept as is

**Audience:**

Yes

**Audience Explanation:**

All reviewers agree that this will be of interest to people in community.

**Claims And Evidence:**

Yes

**Claims Explanation:**

All reviewers agree that claims are supported.